# Deformable Image Registration with Geometry-informed Implicit Neural Representations

**Louis D. van Harten**[1,2]                                      L.D.VANHARTEN@AMSTERDAMUMC.NL
**Rudolf L. M. van Herten**[1,2]
**Jaap Stoker**[3]
**Ivana Išgum**[1,2,3]

[1] *Department of Biomedical Engineering and Physics, Amsterdam University Medical Center - location University of Amsterdam, the Netherlands.*

[2] *Informatics Institute, University of Amsterdam, the Netherlands.*

[3] *Department of Radiology and Nuclear Medicine, Amsterdam University Medical Center - location University of Amsterdam, The Netherlands.*

**Editors:** Accepted for publication at MIDL 2023

## Abstract

Deformable image registration is a crucial component in the analysis of motion in time series. In medical data, the deformation fields are often predictable to a certain degree: the muscles and other tissues causing the motion-of-interest form shapes that may be used as a geometric prior. Using an Implicit Neural Representation to parameterize a deformation field allows the coordinate space to be chosen arbitrarily. We propose to curve this coordinate space around anatomical structures that influence the motion in our time series, yielding a space that is aligned with the expected motion. The geometric information is therefore explicitly encoded into the neural representation, reducing the complexity of the optimized deformation function. We design and evaluate this concept using an abdominal 3D cine-MRI dataset, where the motion of interest is bowel motility. We align the coordinate system of the neural representations with automatically extracted centerlines of the small intestine. We show that explicitly encoding the intestine geometry in the neural representations can improve registration accuracy for bowel loops with active motility when compared to registration using neural representations in the original coordinate system. Additionally, we show that registration accuracy can be further improved using a model that combines a neural representation in image coordinates with a separate neural representation that operates in the proposed tangent coordinate system. This approach may improve the efficiency of deformable registration when describing motion-of-interest that is influenced by the shape of anatomical structures.

**Keywords:** Deformable image registration, geometric prior, cine-MRI, small intestine.

## 1. Introduction

Deformable registration is the practice of transforming multiple images into a shared coordinate space while minimizing the local correspondences of image content. This is an important step in various medical image processing pipelines. It is used for tasks such as joint analysis of multiple modalities (Wells III et al., 1996), longitudinal disease monitoring (Castadot et al., 2010) and motion estimation in cinematic modalities (Wang and Amini, 2011). Deep learning has been widely applied for deformable image registration

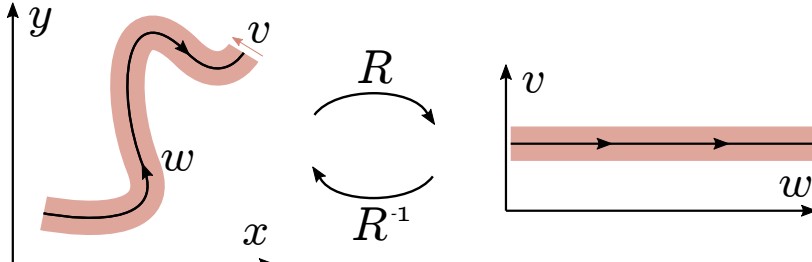

Figure 1: Schematic visualization of transformation $\mathcal{R}$ mapping image coordinate space $xyz$ to tangent space $uvw$. In intestinal motility images, the dominant motion consists of contractions that are aligned with the intestines. Expressing the motion field in terms of tangent coordinates is likely to result in a simpler analytic function than the same motion field expressed in terms of image coordinates, as the latter would need to implicitly encode the curve $w$.

in recent years, but it is has not supplanted classical optimization-based methods: such methods still typically outperform novel learning-based methods in public benchmarks and challenges (Hering et al., 2022). Recent work has proposed the use of Implicit Neural Representations (INRs) for deformable image registration (Wolterink et al., 2022), where the deformation function is parameterized with a lightweight multi-layer perceptron that can be optimized during test-time. This approach outperformed various learning-based methods on the DIR-Lab benchmark.

In the paradigm of registration with INRs, the optimization target is an estimated analytic function that maps $\mathbb{R}^3 \to \mathbb{R}^3$. The complexity of this function determines the required capacity of the neural network and the required computational effort to optimize it, meaning the complexity of this function should be minimized. In the general case with arbitrary sets of input images, it is likely that there is no simpler formulation of the deformation function than the one proposed by (Wolterink et al., 2022), mapping image coordinates to deformation vectors. However, in medical motion analysis tasks, the deformation fields are often predictable to a certain degree: the muscles and other tissues causing and constraining the motion-of-interest form shapes that may be used as a geometric prior. The same is true for various disease development tracking tasks, provided the images from both time-points are aligned to a canonical space. We seek to simplify the optimized deformation functions by exploiting such knowledge of the images to be registered.

Geometric deep learning methods have been proposed to exploit symmetries in data (Bekkers et al., 2018; Weiler et al., 2018). By explicitly encoding these symmetries in a non-parametric fashion, the complexity of an approximated function is reduced. This concept has been extended to gauge-equivariant networks to retain the equivariant structure of convolutions on arbitrary surfaces (Cohen et al., 2019). In this work, we aim to achieve a similar effect in registration of intestinal motility images by aligning our coordinate system with the dominant motion. Rather than embedding geometric structure in our network,

we condition our coordinate system on the centerline curve of the small intestine. We then optimize the deformation function in a tangent space to this curve. A schematic visualization of this transformation is shown in Figure 1. This approach was inspired by work on dynamical system modelling, where aligning local coordinate frames to the motion vectors of nodes in the system has been shown to improve performance (Kofinas et al., 2021).

We compare our proposed method with INR-based registration in image coordinates and analyze the differences in the optimization process and registration performance. We evaluate on an intestinal motility dataset that contains small intestine segments from healthy volunteers, as well as data from inflammatory bowel disease (IBD) patients. A relevant symptom of IBD is reduced intestinal motility (Menys et al., 2018). While some bowel loops in the IBD set express normal motile motion, the average motility in these scans is much lower. Hence, a smaller component of the motion in these cases is aligned with the proposed geometric prior.

## 2. Data

We use an abdominal 4D MRI dataset that contains scans of 14 healthy volunteers, and 10 patients with IBD that were scheduled for ileocecal resection surgery. Each scan consists of volumetric sequences acquired at 1.0 volume per second during a breath-hold. At least 16 timepoints are available for each case. Volumes were acquired at a resolution of 2.5x2.5x2.5 mm and reconstructed to 1.4x1.4x2.5 mm with an FOV of 400x400x35 mm. Small intestine centerline annotations were automatically generated from the first timepoint using the method described by (van Harten et al., 2022). This method yields centerline segments using a neural tracker that terminates either at the border of the field of view, or at the point where the uncertainty of the tracker reaches a threshold. Additional details regarding the centerline extraction can be found in Appendix B. The resulting set contained 117 centerline segments: 67 segments from healthy volunteers and 50 segments from IBD patients.

## 3. Methods

We propose a method for deformable image registration using implicit neural representations conditioned on small intestine centerlines. We explicitly encode the centerline information into the model by transforming the coordinate space of the images into a tangent frame that is aligned with the centerline curve. We compare two versions of our method: one where only the tangent coordinates are input to the model, and one where both the image space and the tangent space coordinates are input to the model.

### 3.1. Implicit neural representations

Implicit Neural Representations are neural networks applied as function approximators, which operate on continuous coordinates rather than image values. The image information only enters the network through backpropagation of gradients from a loss function that relates the coordinates to pixel values.

The method uses neural representation $\Phi(\bar{x}) = \upsilon(\bar{x}) + \bar{x}$ mapping coordinates from source domain $S$ to target domain $T$, where $\upsilon(\bar{x})$ is a sinusoidal representation network

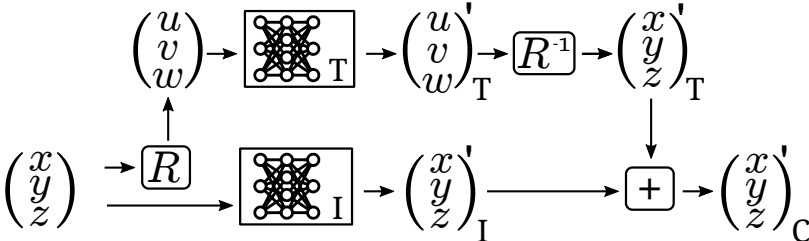

Figure 2: A schematic overview of combined model $\Phi_C$, which uses two networks to operate on both image coordinates and tangent coordinates. $\mathcal{R}$ and $\mathcal{R}^{-1}$ transform samples between the two coordinate systems as shown in Figure 1.

(SIREN) (Sitzmann et al., 2020) that parameterizes the deformation vector field. In contrast to previous work (Wolterink et al., 2022), $\bar{x}$ is not sampled from image space $\mathbb{R}^3$, but from tangent space $uvw$, where $w$ is the curve that describes the small intestine centerline and $u, v$ are basis vectors such that $uvw$ forms a locally Euclidean tangent space around this centerline. Our optimization objective for the tangent space model is:

$$L = L^{data} + \alpha L^{jac}$$
$$= \frac{1}{bs} \sum_{i=1}^{bs} \left( -NCC(\ T[R^{-1}(\bar{x}_i)],\ S[R^{-1}(\Phi(\bar{x}_i))]\ ) + \alpha|1 - \det \nabla \Phi[\bar{x}_i]| \right), \tag{1}$$

where $\alpha$ is a weighting factor, $bs$ is the batch size, $NCC$ is the normalized cross correlation and $R$ is the function that maps tangent coordinates to images coordinates.

Additionally, we propose a combined model that operates on both image space and tangent space coordinates simultaneously. This is achieved by using a separate representation network for each coordinate system and summing the resulting deformation vectors. A schematic visualization of this model is shown Figure 2.

### 3.2. Tangent space definition

Tangent coordinate space $uvw$ is constructed as a parallel transport frame (Hanson and Ma, 1995) around the centerline. Given a smooth centerline, this yields a smoothly varying function of rotation matrices along the centerline curve. To ensure a locally Euclidean space around central axis $w$, we resample each centerline segment as a least-squares polynomial before curve framing. While locally Euclidean, this frame is not an injective function that maps image coordinates to unique coordinates in tangent space: when mapping image coordinates to tangent coordinates, we select the rotation matrix that corresponds to the closest point on the centerline curve.

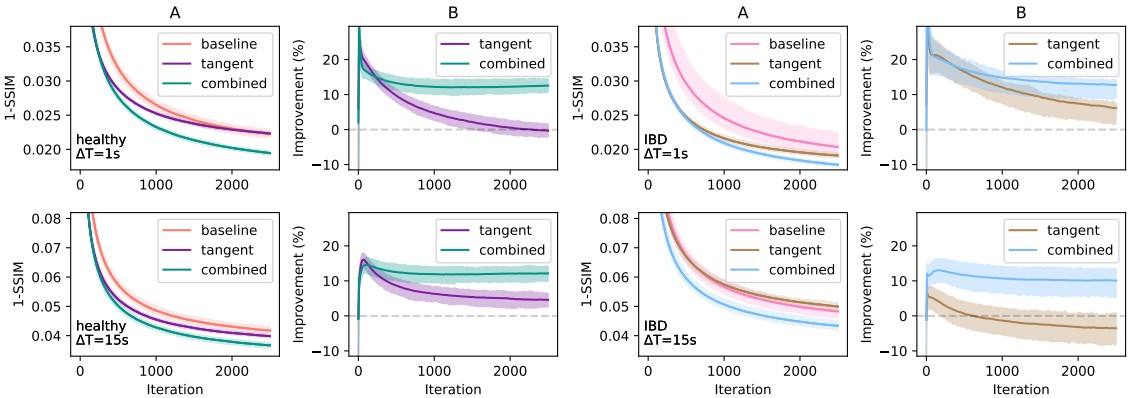

Figure 3: Comparison of the tangent space registration method and the combined tangent space and image space registration method. Results shown for the set of healthy volunteers (left) and the set of IBD patients (right), averaged over all intestine segments for one (top) and fifteen second (bottom) time differences between source and target ($\Delta$T). A: 1-SSIM within the foreground mask after each optimization iteration. B: The average improvement of the proposed methods compared to the image space baseline. Shaded areas indicate 95% confidence intervals.

## 4. Experiments and results

### 4.1. Optimization details

The networks for the INRs are SIRENs that contain 3 hidden layers, initialized with $\omega = 64$. The networks for the baseline and the tangent space model contain 256 neurons per layer. To match the number of parameters in all models, the number of neurons per layer is reduced to 180 in the combined model. The networks are optimized for $2,500$ iterations using the Adam optimizer (Kingma and Ba, 2014) with a learning rate of 1e−4, a batch size of $10,000$ and regularisation weighting factor $\alpha = 0.05$. The foreground mask is defined as a tube around the intestinal centerline with a diameter of 40 mm. This is twice a typical non-contracted small intestine diameter, chosen to account for possible inaccuracies in the extracted centerlines, as well as for possible pathological distension. We center the foreground mask in the coordinate system and we scale the axes to range $[-1, 1]$, constraining the input domain of the function to a single period of the sinusoidal network activation. The code is publicly available[1].

### 4.2. Experiments

We perform pairwise registration for each bowel segment in the dataset, registering the first timepoint to every other timepoint in the sequence. As optimization is a stochastic process, we repeat our experiments with 40 different random seeds to evaluate consistency.

---

1. https://github.com/Louisvh/tangent_INR_registration_3D

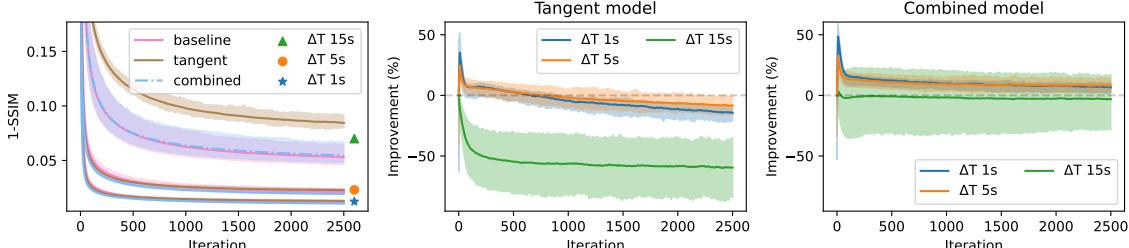

Figure 4: Registration results for a case with breathing motion. Left: The SSIM averaged over the centerline segments from one patient for three time differences ($\Delta$T). Middle: The average relative improvement resulting from the proposed tangent space registration method. Right: The average relative improvement from the proposed combined method. Shaded areas indicate 95% confidence intervals.

To investigate the impact of our centerline prior on the optimization process, we assess the mean absolute error (MAE) and the structural similarity metric (SSIM) between the target image and the transformed source image within the foreground mask after each iteration. First, we compare the optimization performance of the image space registration with the performance of the proposed tangent space registration method and to the combined image space and tangent space registration method for varying differences between the acquisition times of the registered volumes. The results in terms of SSIM are shown in Figure 3. We observe that in all scenarios, the performance in the first 150 iterations is substantially better for both proposed methods. For small time differences, the registration in image space eventually catches up with the tangent space registration method. For large time differences, the average improvement is positive for the healthy volunteers, but *negative* for the IBD patients. Conversely, the combined method that uses both image space coordinates and tangent space coordinates results in a positive improvement in all evaluated scenarios. Quantitative results for other timepoints can be found in Appendix A.

The results in the IBD set are skewed by a number of outliers, caused by cases in which breathing motion is present. The results for the patient with the most severe breathing artifacts are shown in Figure 4. For this patient, the registration results from the proposed method are similar to the results from the baseline method for time differences of 1 and 5 seconds. However, the tangent space results are approximately 20% worse than the baseline when the method is applied to the timepoint in which the patient lost control of their breath.

Several qualitative results are shown in Figure 5, comparing the properties of the image space and the tangent space models for three different scenarios. The first are cases where the tangent space registration is beneficial, as shown in 5.A. In these cases, the dominant motion in the bowel loop is caused by its motility, resulting in a deformation function that is easier to express in tangent coordinates. The second are cases where the tangent space method is not beneficial, either due to an incorrectly extracted centerline or due to an absence of motility, as shown in 5.B. The third are cases where aligning the coordinate system with the intestinal centerline is actively harmful to the registration result. This

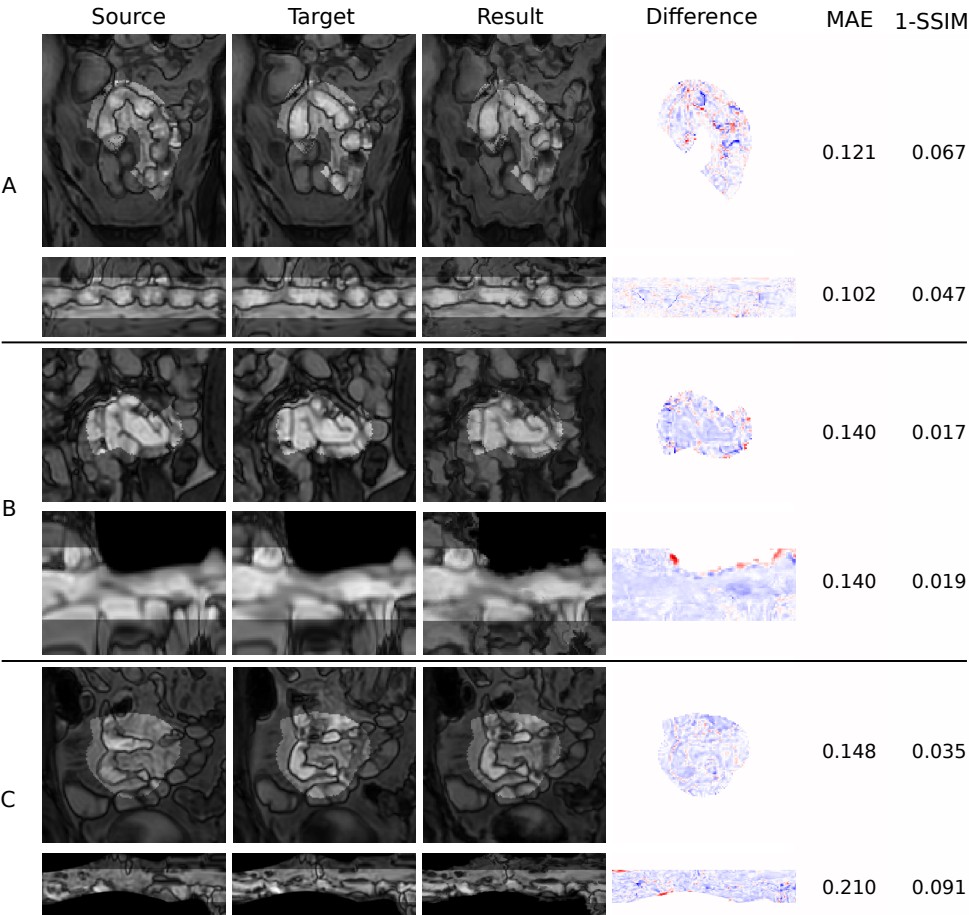

Figure 5: Three examples of registration results from the image space registration (top) and the proposed tangent space registration (bottom). A) Registration with geometric prior is beneficial (16% MAE improvement); the centerline is correctly positioned in both timepoints and there is motile activity. B) The prior is not beneficial (–0.4% MAE improvement); no motile activity is present in the bowel loop. C) The prior is harmful (–42% MAE improvement); breathing motion moved the bowel loop away from the centerline curve in one of the timepoints.

includes cases where the subject is unable to maintain the breath-hold, such that breathing becomes the dominant motion (as shown in 5.C).

## 5. Discussion and Conclusion

We have presented a method for deformable image registration with implicit neural representations that incorporate knowledge of the physical geometry in the optimization process. By aligning the coordinate system of our problem space with small intestine centerline

curves, we construct a locally Euclidean tangent space to simplify the estimated function that describes the intestinal motility. In all of our experiments, the proposed tangent space registration method resulted in the average mean absolute error and SSIM improving faster than the image space baseline. This suggests that the functions estimated in tangent space are indeed simpler to optimize than their counterparts in the original coordinate system.

Our proposed tangent space registration method outperforms the image space baseline in bowel segments where motility is present, but has no benefit in dysmotile bowel segments. This is in line with intuition: the geometric information simplifies the function describing the local motion, which is aligned with the centerline curve. Motion from other sources, such as nearby bowel segments, is not aligned with this same curve. In cases where breathing motion is present, the proposed tangent space registration resulted in a performance reduction. Breathing motion can be captured in a simple mathematical function when expressed in image coordinates, as all voxels of interest move in approximately the same direction. Conversely, in tangent coordinates, this function has to implicitly encode the centerline curve to capture the same motion. This effect could explain the success of the combined method, which can implicitly decompose the deformation fields into components that are more simple to express in either image space or tangent space coordinates.

In registration of timepoints with small time differences, the tangent space method attains adequate registration results faster than the baseline. However, these differences disappear as the models are optimized for more iterations. We initially assumed that this happens because the registration problem for small time differences is sufficiently simple that explicitly providing the shape of centerline curve $w$ has no benefit over implicitly learning this same curve. However, as the combined method does result in better performance for this setting, this assumption seems to be invalid. Instead, this effect may have been caused by the chosen evaluation strategy. We evaluate the registration results within a tubular foreground mask with twice the diameter of a typical non-contracted small intestine. Hence, part of the evaluated voxels picture adjacent bowel segments and other abdominal tissues. The deformation of the tissues outside of the central bowel segment are generally likely simpler to express in image coordinates than in tangent coordinates.

The tangent space used in this work is common to the source and target images. It is constructed from a centerline that was automatically extracted in the first timepoint. This relies on the assumption that the abdominal centerlines have little to no movement in position and shape throughout the breath-hold. In many of our images, this assumption is violated to some degree. Centerlines drift away from their original positions due to the natural movement of the bowels, as well as due to clenching muscles when subjects have trouble holding their breath. This effect could be avoided by using separate centerlines in each timepoint. However, if there are any geometric differences (e.g. if either centerline is slightly off-center in any curved section), the resulting tangent spaces in the source and target domains would be inconsistent, with mismatched relative rotations of the normal planes. Hence, this approach would only be feasible with very high quality centerline annotations, which is challenging to ensure. The resulting quality control burden would make clinical application infeasible. However, as shown in our experiments, it is possible to circumvent the limitations of using a single centerline by combining the tangent space with image space information.

Reducing the complexity of a deformation function by expressing the problem in a more suitable coordinate system is not constrained to parallel transport frames around centerlines. The same concept could be applied to any motion estimation problem where a component of the expected motion is aligned with a known curve or surface. For example, this concept could be applied in the description of cardiac motion, where a coordinate system aligned with a myocardium boundary may be beneficial, or in ischemic stroke follow-up, where a spherical coordinate system could be aligned with the center of the stroke lesion.

In conclusion, this work has presented a deformable registration method that uses geometry-aware implicit neural representations. These representations are explicitly conditioned on the shapes of anatomical structures that influence the expected motion. Our experiments show that this method may improve the efficiency of deformable registration when describing motion-of-interest that is influenced by known anatomical shapes.

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

## Appendix A. Additional results

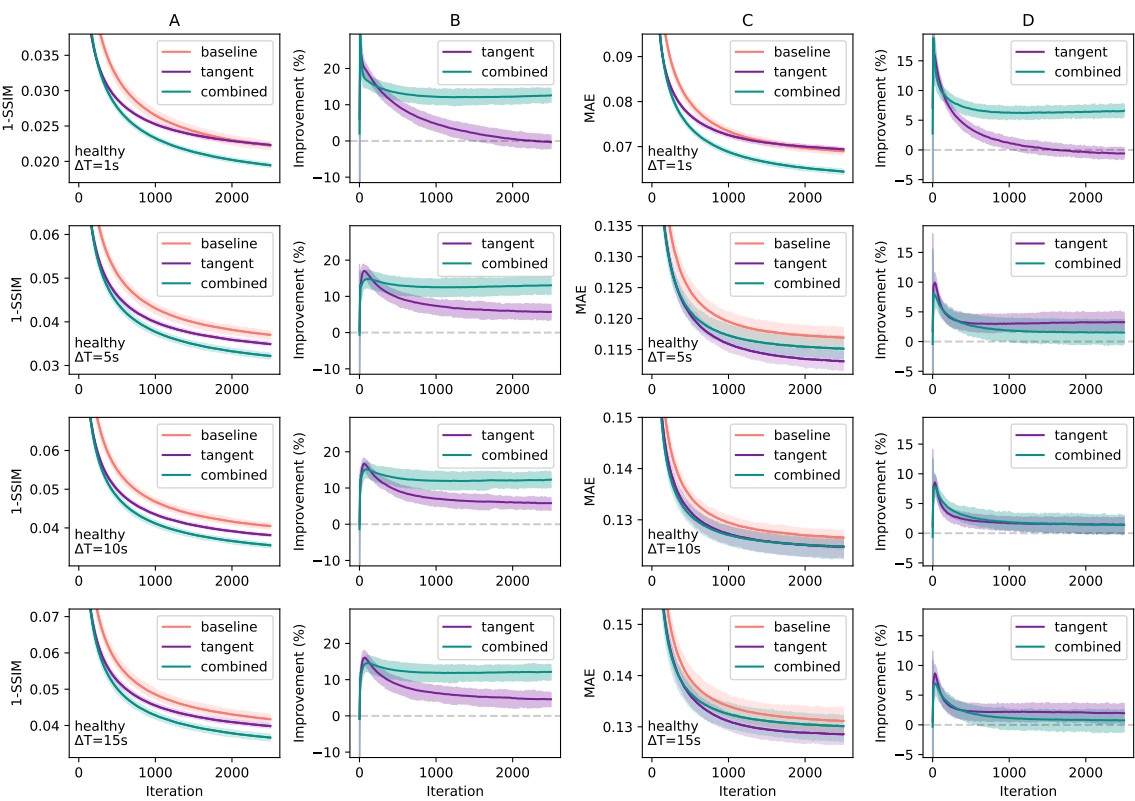

Figure 6: Results for the set of healthy volunteers, averaged over all intestine segments for increasing time differences between source and target ($\Delta$T). A: 1-SSIM within the foreground mask after each optimization iteration. B,D: The average improvement of the proposed methods compared to the image space baseline. C: MAE within the foreground mask after each optimization iteration. Shaded areas indicate 95% confidence intervals.

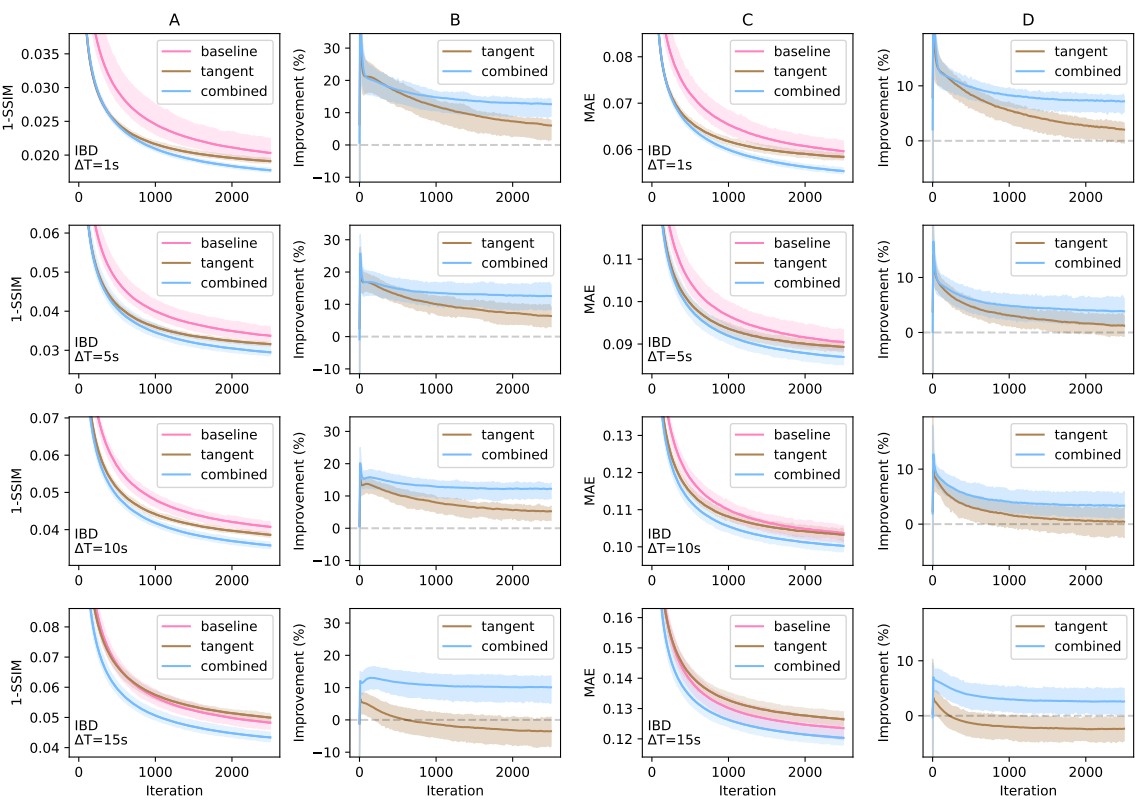

Figure 7: Results for the set of IBD patients, averaged over all intestine segments for increasing time differences between source and target (ΔT). A: 1-SSIM within the foreground mask after each optimization iteration. B,D: The average improvement of the proposed methods compared to the image space baseline. C: MAE within the foreground mask after each optimization iteration. Shaded areas indicate 95% confidence intervals.

## Appendix B. Supplementary dataset details

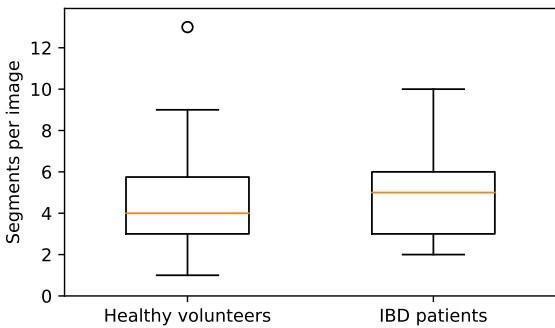

Figure 8: Number of selected centerline segments per image in each set.

In addition to image volumes, the presented method requires a geometric prior to construct a tangent coordinate space. For the experiments presented in this work, we construct this tangent space around automatically extracted centerlines of small intestine segments. Centerline segments were extracted from manually placed seed points using the method described in (van Harten et al., 2022). This method yields centerline segments using a neural tracker that terminates either at the border of the field of view, or at the point where the uncertainty of the tracker reaches a threshold. Performance for this method in healthy volunteers and IBD patients has been reported as similar.

Only segments of at least 10cm were selected for analysis, as short segments are unlikely to be relevant in downstream motility analysis tasks. The extracted segments may partially overlap due to tracking mistakes; in cases where multiple extracted centerline segments overlap more than 33%, the shortest segment was discarded. An overview of the number of selected segments per image is shown in Figure 8.

