# OpenReview forum: "Deformable Image Registration with Geometry-informed Implicit Neural Representations"
_MIDL.io/2023/Conference — MIDL 2023 Poster_

### Official Review · Reviewer_gs53 · 2023-02-02

**Confidence:** 3
**Preliminary Rating:** 4
**Recommendation:** Poster

**Summary:**

The authors introduce a method for image registration for a specific type of image for which a meaningful centerline can be estimated, such as intestines. They see centerline-based coordinates instead of standard image coordinates and show increased performances of the resulting registration algorithm on a dataset of intestine images.

**Strengths:**

The paper is well-written and clear, the methodology is easy to follow and the results are well-presented. The proposed method clearly improves the registration when the image deformations are local and it is well-explained why it does not work for global deformations.

**Weaknesses:**

I see two weaknesses. First, the method does not work well on global deformations, which I think could be probably easily solved (see below). The second is on the scope of applicability of this method. Apart from intestine data, for what other types of images would this method be appropriate?

**Deanonymize Review:**

no

**Detailed Comments:**

I don't have any.

**Paper Type:**

both

**Questions To Address In The Rebuttal:**

Regarding the loss of accuracy on global deformation, what would be possible fixes? I would suggest proposing some in the discussion section. You mention that an additional simpler rigid registration could be applied to the images, but would a  more subtle combination of position and tangent vectors in the algorithm work?
I would also be interested to have more on the applicability of this method for other types of datasets, to broaden its scope.

---

### Official Review · Reviewer_nF6V · 2023-02-03

**Confidence:** 4
**Preliminary Rating:** 3
**Recommendation:** Poster

**Summary:**

This paper proposes a variation of a recently introduced Implicit Neural Representation (INR) based image registration method, where the coordinates mapped by the network representing the deformation vector field (DVF) are expressed in a local reference system aligned along the small intestine centerlines. Such system has the advantage of naturally capturing the main trajectories of the subject motion in the data used for the experiments in the paper.

**Strengths:**

This paper builds upon the INR-based registration method recently introduced in (Wolterink at al. 2022) and introduces a local coordinates system aligned along the centerlines of the small intestine for expressing the coordinates transformation. Such method has the advantage of allowing for faster convergence of the registration process, suggesting that changing to a more natural frame or reference for the problem at hand does simplify the representation of the coordinate mapping. The method is validated against a set of healthy and pathological data, showing that when the data content actually satisfy the geometrical prior, registration is more efficient. The paper is well written and rather easy to follow, the general ideas are well outlined. The proposed technique is novel and might be beneficial in other areas as well, like pulmonary airways or vessels trees, and as such be of general interest for anatomic specific applications.

**Weaknesses:**

* The soundness of the method relies on two key factors, (1) the quality of the centerline extraction method for the definition of the local coordinate system, and (2) whether the subject motion actually occurs along the hypothesized local coordinates.
* (1) a more thorough assessment of the centerline extraction performance is needed, which is key for the soundness of the evaluation pipeline.
* When (2) is not satisfied, the authors show that no improvement against the baseline is reported. This raises concern about the genericity of such approach in the clinical practice.
* The preparation/handling of the data and explanations of the methods are not always clear nor complete.

**Deanonymize Review:**

no

**Detailed Comments:**

1. Missing citation for SIREN.
2. Figure 2. Please explain in the caption what are the Delta Ts reported in the figure legend.
3. In Section 2 (Data), it is stated in the last period that 67 segments are extracted from 14 healthy volunteers and 50 segments from 10 IBD patients. Without further context (inferable later), this information is confusing. I'd anticipate that registration pairs are the mentioned bowel segments per frame instead than the entire time frame images.
4. In Section 2 (Data), it is not clear whether the centerlines are extracted in each time frame of the 4D MRI or only on the first one. Reading caption of Fig. 3 "C) The prior is harmful (−42% MAE improvement); breathing motion moved the bowel loop away from the centerline curve in one of the timepoints" I understand that the centerline is extracted on the first frame, and this is only confirmed in the conclusions when the authors say "The tangent space used in this work is common to the source and target images. It is constructed from a centerline that was automatically extracted in the first timepoint.". This should be much more clearly addressed in the Data section.
5. Again in Section 2 (Data) it would be helpful to introduce a small table on how many segments per image were extracted, together with an assessment of the centerline extraction method performances. Furthermore, is the quality of the extraction affected by the imaging subject being healthy/pathological? Are there any bias in the centerline because of the presence of IBD?
6. Not clear how the MAE introduced in (Section 4.2 Experiments) is defined. Is the ground-truth DVF known? Is it derived in term of of the centerlines? If yes, how? This must be specified.
7. At page 5 the sentence starting with "Several qualitative results are shown in Figure 3." should go into a subsection for ease of reading. Also first explain why you're splitting the data into subgroups (i.e. in order to evaluate when expressing the registration process in terms of geometric priors is beneficial), and then describe the experiment in detail.
8. Page 7, typo "qualitatively inspection" -> "qualitative inspection", "this" repeated twice.
9. I'm not a clinical expert, but the fact that the baseline registration method performs better for large point distances might also be due to periodicity effects in the bowel movement, such that the better performance is simply "good luck".
10. When Dt > 1, it is not clear to me if time frame 1 is directly registered against frame N, or the registration occurs iteratively for each successive frame, i.e Reg(1, 2) -> Reg(2,3) -> ... -> R(N-1, N). Please clarify in the method section.
11. Since performance of the proposed method is not significantly better than the baseline when registering successive time frames, but it is better for distant time frames, it would be very interesting to assess the performance of the two methods when iteratively registering each time frame to the previous one until the last frame available (If that's not what is done: see point 10).
12. It is not clear why the centerlines are not extracted for each time frame. If the centerlines were available for each time frame, why not rigidly pre-aligning the segments along the centerlines as a first step?

**Paper Type:**

methodological development

**Questions To Address In The Rebuttal:**

The following points should be addressed or better explained:
* Quality of the centerline extraction, potential biases and how this could affect the registration performance.
* Clarify why the centerline extraction is performed only on the first frame.
* Exact definition of the MAE. If, as I understand, the MAE is computed in term of centerlines, the evaluation could be potentially flawed due to the accuracy of the extraction method. It would be beneficial to manually set landmarks in the images and evaluate against that. If that's prohibitive in term of annotation effort, I'd like to read an assessment of that.
* Please clarify whether for time points larger than 1s the registration is performed directly between time 0 and time N, or it is iterated through all the intermediate time points.

---

### Official Review · Reviewer_d4U4 · 2023-02-03

**Confidence:** 3
**Preliminary Rating:** 4
**Recommendation:** Poster

**Summary:**

The authors propose a method for deformable image registration, that incorporates a geometric prior (i.e. the intestine centerline) into implicit neural representations (INR). To achieve this, they transfer the image space to a tangent space around the centerline before applying INR. They showed that their method outperforms image space INR for registering bowel motility, but is disadvantageous for images with breathing artifacts.

**Strengths:**

- The authors propose an adaption to implicit neural representations by firstly mapping the image space to an anatomically relevant tangent space.
- The proposed methods outperforms the baseline for scans of healthy patients.
- For failed cases, the authors pinpointed why their method was outperformed by the baseline.
- The paper is well written and easy to follow.

**Weaknesses:**

- For IBD patients, the disadvantages seem to outweigh the advantages of the proposed method, resulting in an overall worse registration accuracy.
- The authors do not provide a comparison to other registration methods incorporating geometric priors.
- In their discussion, the authors propose the use of a rigid pre-registration to address breathing artefacts. The paper could benefit from showing how strongly this influences the accuracy.

**Deanonymize Review:**

yes

**Detailed Comments:**

„Qualitatively inspection of the results reveals this this…“ > typo

**Paper Type:**

methodological development

**Questions To Address In The Rebuttal:**

- Did the authors do an experiment incorporating rigid pre-registration? If so, how did their method perform on images with breathing artifacts?
- How do the authors view the clinical applicability of their methods considering the disadvantage of their method for IBD patients?

Final justification
The authors have appropriately answered my questions and also in quite some detail the other concerns of fellow reviewers, I hence stick to my initial rating of "weak accept"

---

### Meta-Review · Area_Chair_wa58 · 2023-02-23

**Recommendation:** Accept (Poster)
**Confidence:** 5

**Metareview:**

This paper proposes a deformable registration model which uses Implicit Neural Representation to parameterize the deformation field and provide experimental validation in the context of abdominal 4D MR. Two reviewers recommend weak acceptance of the paper, while one reviewer assigned Borderline. After reading the rebuttal provided by the authors, I believe most of the reviewer’s comments have been addressed. Thus, I think this paper can be accepted for publication at MIDL.